# Ultrasound-Assisted Extraction of High-Value Fractions from Fruit Industrial Processing Waste

**DOI:** 10.3390/foods11142089

**Published:** 2022-07-14

**Authors:** Rebeca Esteban-Lustres, Vanesa Sanz, Herminia Domínguez, María Dolores Torres

**Affiliations:** CINBIO, Department of Chemical Engineering, Faculty of Sciences, Universidade de Vigo (Campus Ourense), Edificio Politécnico, As Lagoas, 32004 Ourense, Spain; rebeca.esteban@uvigo.es (R.E.-L.); vsanz@uvigo.es (V.S.); herminia@uvigo.es (H.D.)

**Keywords:** antioxidants, protein, probe sonication, green technologies, food disposals

## Abstract

This work deals with the valorization of fruit industrial processing waste pretreated with two dehydration methods, air oven and lyophilization. Ultrasound-assisted extraction using a sonication probe was selected to recover the high-value fractions. A battery of experiments following a Box–Behnken design was planned to evaluate the effect of the ultrasound amplitude, extraction duration, and temperature on the yield, protein content, phenolic content, and antiradical capacity of the soluble extracts. Operating at a fixed frequency (24 kHz) and solid:water ratio (1:15), the models predicted (significance degree >95%) the maximum extraction conditions of 69.7% amplitude, 53.43 °C, and 12 min for conventionally dehydrated fruit waste. Under these processing conditions, 52.6% extraction yield was achieved, with a protein content of 0.42 mg/g, total phenolic content of 116.42 mg GAE/g, and antioxidant capacity of 44.95 mg Trolox/g. Similar yields (53.95%) and a notably higher protein content (0.69 mg/g), total phenolic content (135.32 mg GAE/g), and antioxidant capacity (49.52 mg Trolox/g) were identified for lyophilized fruit waste. This treatment required a longer dehydration pretreatment duration (double), higher ultrasound amplitude (80%), and higher extraction temperature (70 °C), but shorter extraction time (4 min). These outcomes highlighted the important impact of the dehydration method on the valorization of the tested waste, with conventional drying saving costs, but the lyophilization procedure enhancing the bioactive features of the waste.

## 1. Introduction

Currently, the agri-food sector faces two important challenges: the recovery of waste from the various production and logistical stages, and its use as an initial raw material for the synthesis of new products with greater added value [1,2]. According to the Food and Agriculture Organization of the United Nations, disposals in the fruit and vegetable industries are the highest among all types of food and can reach up to 60% of the overall production [3]. The fruit by-products resulting from the transformation processes of the agri-food industries are considered an alternative sustainable and renewable raw material with low economic value [4]. Their use for the production of high-added-value products, including biofertilizers, dietary fibers, industrial enzymes, and biogas, as well as their application as substrates for the production of several valuable bioactive compound fractions (phenolic compounds, vitamins, polysaccharides, essential oils, among others), has caused great scientific interest in recent years [5].

Phenolic compounds in the daily diet can contribute to the prevention of cardiovascular and neurodegenerative diseases, osteoporosis, cancer, and diabetes mellitus [6], with their content and antioxidant capacity being higher in the peel and seeds of fruits than in the pulp [7]. Phenolic contents up to 15% higher were reported for the peels of lemons, grapes, and oranges and the seeds of avocados and mangos [4]; up to three times higher for banana skins [4]; and up to 15 times higher for kiwi peels [8] when compared with the corresponding pulps. The non-edible portions and residues of fruits are also interesting protein sources, ranging from 2.5 to 9.0% for citrus peel [4], from 2.7 to 5.3% for apple skin [9], up to 34% for apple seeds [10], around 6% for banana peel [4], about 8% for pineapple skin, and 15% for melon skin [11].

Due to their high moisture content (80–90%) and acidic pH, fruit by-products are perishable and difficult to store, and so it is necessary to apply a pretreatment prior to the bioactive compound extraction [12]. The selection of an adequate dehydration treatment method under the most favorable conditions is very important to extend the useful life of the waste without jeopardizing its bioactive features, since the drying of fruit by-products can reduce their antioxidant properties [13]. The subsequent selection of the extraction method is another key factor for achieving high yields, preventing the degradation of compounds, reducing the use of organic solvents, and saving time and energy [14]. An attractive environmentally friendly alternative could be ultrasound-assisted extraction using an ultrasonic probe under optimized processing conditions (e.g., solid-liquid ratio, temperature, time, and power) [15]. Several studies have been published on the application of ultrasound-assisted extraction for the recovery of antioxidant compounds from individual fruit by-products using both ultrasonic baths and probes [15,16,17]. However, further knowledge is needed regarding the global valorization of mixed fruit processing residues, where different fractions may be involved. Iqbal et al. [16] found a 30% increase in the yield of total phenolic compounds in orange peel when compared with conventional extraction, and a 65% rise in the yield from grape pomace when compared with maceration. Kumar et al. [17] also summarized in a comprehensive review results showing an increase in the total phenolic compounds and antioxidant activities of mango and pomegranate peels and coconut shells after conventional ultrasound treatment.

Fruit processing operations produce significant by-product waste, which can create serious nutritional, economic, and environmental drawbacks. In this context, the main aim of this work is the valorization of fruit processing industrial waste through the optimization of an environmentally friendly extraction process to obtain high-value extracts, assessing the impact of the dehydration pretreatment of the waste as a key factor in the processing.

## 2. Materials and Methods

### 2.1. Materials

Fruit industrial processing waste used as a raw material was kindly provided from a local company (Galifresh, Pontevedra, Spain). The residues were predominantly peels (≈55%), seeds (<5%), peduncles (<5%), and the remains of the pulp (≈35%) of oranges, bananas, pears, and apples in variable proportions. All samples were frozen at −18 °C in airtight plastic bags before dehydration treatment.

### 2.2. Dehydration Treatment

Two dehydration methods were tested in order to determine their impact on the bioactive properties of the recovered extracts from fruit industrial processing waste. Conventional drying was conducted in a convective air oven (P-Selecta, Barcelona, Spain) at 60 °C until constant weight following the corresponding drying kinetics, and samples were labelled FE. Freeze-drying was carried out in a freeze-drier (Alpha 2–5 LD plus, Martin Christ, Germany) operating at −55 °C, 0.021 mbar, and samples were labelled FL. In both cases, micronized samples were stored at room temperature in the dark in airtight plastic bags until further analysis. The drying kinetics of both processes were determined as moisture ratio (MR = (M_t_ − M_e_)/(M_0_ − M_e_)) versus time, with M_0_ and M_e_ being the initial and equilibrium moisture content (kg water/kg dry basis, d.b.), respectively, and M_t_ the moisture content at any dehydration time. The corresponding drying kinetics were modeled using the well-known Page model [18] (${\mathrm{MR}=e}^{-{\mathrm{kt}}^{n}}$), where k and n are the two model parameters.

It should be highlighted that the energy consumption (E) for both dehydration treatments was estimated as the multiplication of the required power (P) by the dehydration time (t) [19].

### 2.3. Fundamental Chemical Characterization

Moisture content of fruit waste dehydrated by both methods was determined at 105 °C using a standard gravimetric method, 964.22 [20].

Ash content was gravimetrically estimated by calcinating the samples at 575 °C in an ELF 11/6B muffle furnace (Carbolite Gero Limited, England). The macro- (Na, K, Mg, Ca) and micro- (Cu, Zn, Fe, Cd, Hg, Pb) elements were measured by inductively coupled plasma optical emission spectrometry (Optima 4300 DV, PerkinElmer, Massachusetts, MA, USA), except for Hg content, which was determined using cold vapor atomic absorption spectrometry.

Total nitrogen content was quantified by the Kjeldahl method, and the corresponding total protein content was estimated by multiplying the nitrogen content by a factor of 6.25, as commonly used for foodstuff [21].

The carbon content of the dehydrated samples was measured by flash combustion technology employing a gas chromatographic EA 1112 analyzer with a thermal conductivity detector (Thermo Flash, Berlin, Germany), following the procedure previously detailed in [22]. Carbohydrate profiles were determined by means of high-performance liquid chromatography (HPLC) with a previous acid hydrolysis treatment [22]. Concisely, samples were initially treated at 30 °C with sulfuric acid (72% (*v*/*v*)) for 60 min, followed by a treatment at a lower solvent content (4% (*v*/*v*)) at 121 °C for 60 min. The obtained liquid phase was filtered to recover the non-hydrolyzed material in order to assess the acid-insoluble residue after drying at 105 °C. The corresponding liquid extract was analyzed by means of HPLC using an HPX-87H Aminex column (Bio-Rad Laboratories, Barcelona, Spain) operating at 50 °C with sulfuric acid (3 mM) as mobile phase (flow rate of 0.6 mL/min) and a refractive index detector (Agilent Technologies, California, CA, USA).

### 2.4. Ultrasound-Assisted Extraction

Fruit waste dehydrated by both proposed treatments was subjected to ultrasound-assisted extraction using a sonication probe of 22 mm (LABSONIC P, Sartorius Stedim, Berlin, Germany) operating at 24 kHz and a maximum power of 400 W. Temperature control of the jacketed glass beaker, where the samples were placed, was monitored within a closed circuit using a cryostatic bath. The set point temperature for each extraction was programmed in the bath and controlled by a thermometer inside of the beaker. Dried samples were mixed with distilled water at a fixed solid:liquid ratio (1:15, *w*/*w*) within the jacketed glass beaker and placed on the soundproofing box, with the probe located on the central axis of the reactor, under different processing conditions defined by the Box–Behnken design. The recovered extract was vacuum filtered and centrifuged (Hettich Rotixa 50 Rs, Berlin, Germany) at 3500 rpm for 20 min at 15 °C. The separated extracts were frozen at −18 °C until further analysis.

### 2.5. Box–Behnken Experimental Design

The combined impact of three independent variables on the recovery of high-value compounds from fruit industrial processing waste was assessed using a Box–Behnken design (Table 1), as previously reported elsewhere [23]. The corresponding nomenclature and value ranges were as follows: amplitude (A, 20–80%), extraction temperature (T, 30–70 °C), and extraction time (t, 4–12 min). Several preliminary trials were performed not only to define the independent variables ranges, but also to stablish the fixed operation conditions, such as the solid:liquid ratio (1:15, *w*/*w*), the extraction cycle (0.5 s), and the ultrasound frequency (24 kHz). Note here that the proposed independent variable ranges were set by taking into account a compromise between extraction speed, yield, and thermal degradation. The measured responses (dependent variables) selected to assess the potential of the soluble extracts of both FE and FL samples were the extraction yield (g/100 g dry fruit waste), total phenolic content (mg GAE/g), antioxidant capacity (mg Trolox/g), and protein content (g BSA/100 g).

The experimental design created using Minitab 21 software (Minitab LCC, Pennsylvania, PA, USA) consisted of 15 combinations, with the response function (Yi) separated into linear, quadratic, and interactive terms:$$Y_{i}=a_{0}+a_{1}X_{1}+a_{2}X_{2}+a_{3}X_{3}+a_{4}X_{1}X_{2}+a_{5}X_{1}X_{3}+a_{6}X_{2}X_{3}+a_{7}X_{1}{}^{2}+a_{8}X_{2}{}^{2}+a_{9}X_{3}{}^{2}$$
where *X*_1_–*X*_3_ are the independent variables; *a*_0_ the intercept of the model; and *a*_1_–*a*_3_, *a*_4_–*a*_6_, and *a*_7_–*a*_9_ the coefficients of the linear, interactive, and quadratic effects, respectively.

### 2.6. Soluble Extract Properties

The extraction yield was gravimetrically measured as the g extract/100 g dried fruit waste, as reported elsewhere [17].

The total phenolic content of the soluble extracts obtained after the sonication treatment of FE and FL samples, expressed as gallic acid equivalent (GAE), was determined using the Folin–Ciocalteu method [24]. Briefly, the liquid extract (0.25 mL) was mixed with distillated water (1.9 mL), the Folin reagent (0.125 mL), and sodium carbonate (10%, 0.25 mL). Samples were homogenized in a vortex and kept at room temperature in the dark for 60 min before measuring the absorbance at 765 nm.

The antioxidant activity of the above extracts, estimated as ABTS radical cation scavenging, was assessed following the method detailed by Re et al. [25]. Briefly, ABTS (1 mL) was mixed with the recovered soluble extracts (10 µL) or Trolox reagent in a water bath at 30 °C. Absorbances were read at 734 nm at intervals of 6 min after ABTS reagent addition, and the results were expressed as Trolox equivalent antioxidant capacity (TEAC).

The protein content of the extracts (FL, FE) was spectroscopically measured using the well-known Bradford protocol [26]. Concisely, Bradford reagent (0.5 mL) was mixed with the tested extracts (0.5 mL) or bovine serum albumin. In all cases, the absorbance was measured at 595 nm after 20 min of incubation at room temperature.

All above analytical measurements were conducted at least in triplicate.

### 2.7. Statistical Analysis

Statistical analysis was carried out by means of three-factor analysis of variance employing Minitab 21 software (Minitab LCC, Pennsylvania, PA, USA). The regression coefficients of individual linear, quadratic, and interactive terms of the Box–Behnken experimental design and their corresponding effects were assessed. An F-value with 95% confidence (*p* < 0.05) was calculated to evaluate the significant terms in the polynomial.

## 3. Results and Discussion

The optimized ultrasound-assisted extraction of fruit industrial processing waste previously subjected to two dehydration treatments was carried out following the general overview displayed in Figure 1. The impact of the extraction temperature and time as well as the ultrasound amplitude was successfully evaluated according to a Box– Behnken experimental design. The separated soluble fractions were individually and jointly evaluated for yield, protein, total phenolic content, and antioxidant capacity with promising results.

### 3.1. Dehydration Kinetics

Fruit industrial processing residues presented an initial moisture content of 72.9 ± 1.2% and ash content of 0.67 ± 0.10%. Due to their high humidity values, these residues are perishable and generate problems for storage and the environment, so an adequate pretreatment is necessary for their subsequent use in the recovery of the target compounds [12]. Figure 2 presents the drying kinetics determined for fruit industrial processing waste after conventional processing and lyophilization, with the corresponding models. In both cases, the water removal was more pronounced at the beginning of the drying process (the free water release stage), since the loss of surface water from the solid caused water to diffuse from inside the solid to its surface. When the moisture concentration decreased, the rate of drying also decreased. A significant amount of time was saved for samples dehydrated in the convective air oven when compared to those placed in the freeze-drier, which took around 20 and 40 h, respectively, to achieve final moisture content values below 10%, ensuring sample preservation. In both cases, the dehydration kinetics were adequately (R^2^ > 0.92, root mean square error (RMSE) < 0.05) fitted using the Page model. Samples conventionally dehydrated exhibited higher k and lower n values (k: 0.62 h-n; n: 0.50) when compared to the lyophilized samples (k: 0.12 h-n; n: 0.88). These values agree with those found for other fruit wastes with similar initial moisture contents dehydrated by conventional [27] and emerging [28] treatments.

According to the estimation of the energy consumption, oven drying by convective air was the most efficient method from the energy point of view, since it consumed less energy (FE: 28 kJ, FL: 36 kJ) and took less time to dry the samples while maintaining their stability.

### 3.2. Fundamental Composition of the Dehydrated Fruit Waste

Table 2 summarizes the effect of the drying treatment (i.e., FE and FL) on the composition of the fruit waste, with statistical differences evident in some fractions. FE exhibited a slightly higher moisture content (about 8%) when compared with FL (about 6%), though in both cases the value was lower than 10%, which guarantees dehydration, inhibits microbial growth, and increases the stability of the samples [29]. Similar values were identified for the ash content, which was consistent with those previously reported for lime skins (5.84 ± 0.39%) and eggplants (9.21 ± 0.21) [21]. The average protein content values (in dry weight) varied between 6.32% for FE and 5.34% for FL, agreeing with those obtained for apple pomace (2.7–5.3%) [9], banana peel (6.02%), and orange peel (5.97–6.4%) [4,30]. The largest differences in mineral content between FE and FL were identified for sodium, suggesting that smaller ions can be more easily entrained during the lyophilization process. Concerning carbohydrate content, no significant differences were observed between fruit waste subjected to different dehydration treatments, with values around 50%. This value is greater than the content range found for fruit residues such as bananas and cocoa (47.68 and 47.78%, respectively) [31]. The monosaccharide composition indicated that glucose was the main sugar in the tested industrial fruit residues, followed by xylose and arabinose. The corresponding values were 26.7 ± 0.35%, 8.53 ± 0.06%, and 3.57 ± 0.04% for FE and 27.68 ± 0.18%, 8.24 ± 0.05%, and 3.51 ± 0.02% for FL, respectively. The carbohydrate and protein content values make this biowaste an attractive source for fermentation substrates of organic origin [20].

As expected, the tested fruit waste was a rich source of minerals, with particularly high potassium, calcium, and magnesium content. The FE samples showed statistically higher values of these compounds. These minerals can neutralize the acidic effects of low-pH foods in the diet [32]. It is noteworthy that the calcium content of the tested fruit waste (491.1 and 434.5 mg/100 g for FE and FL, respectively) was in the range of that found for 100 g of soft cheese, which contains an average of 470 mg of calcium according to the Spanish Food Composition Database [33]. This behavior suggests that the tested waste could be used for food fortification. The magnesium values of the tested waste were comparable to those of recognized vegetable sources of magnesium, such as chard (86 mg/100 g) and spinach (63 mg/100 g). The iron content was also consistent with the values reported for meat (2–4 mg/100 g) [6], suggesting the potential use of these residues in the development of mineral-enriched foods, as long as it is ensured that the bioavailability meets dietary requirements.

### 3.3. Soluble Extract Properties

Figure 3 shows the main characteristics determined experimentally for the soluble extracts obtained after the sonication treatment of both the FE and FL samples under the operation conditions established in the Box–Behnken design. Note here that the dry content of all tested extracts varied within a narrow range (0.034 ± 0.003 g/g extract). The extraction yields exhibited similar tendencies for the FE and FL systems (Figure 3a). In both cases, the highest yield (FE: 54.72 ± 2.28%, FL: 54.55 ± 0.08%) was identified for sample 1 (50% amplitude, 70 °C, 12 min), without notable differences between extractions 2, 3, and 5. The lowest yield (FE: 46.19 ± 0.15%, FL: 46.51 ± 0.18%) was found for sample 12 (50% amplitude, 30˚C, 4 min). These results suggest that at a constant amplitude (50%), an increase in the temperature and duration improved the extraction yield percentage in both cases.

The protein content of the corresponding soluble extracts is displayed in Figure 3b, varying from 0.21 ± 0.01 to 0.46 ± 0.01 g/100 g for FE and from 0.19 ± 0.01 to 0.67 ± 0.01 g/100 g for FL. The maximum value for FE was obtained for samples 1 (50% amplitude, 70 °C, 12 min) and 7 (80% amplitude, 50 °C, and 4 min), whereas the minimum value was identified for sample 10 (20%, 30 °C, 8 min). Concerning the FL extracts, the highest value was found for sample 2 (80% amplitude, 70 °C, 12 min) and the lowest for sample 12 (50% amplitude, 30 °C, 4 min). These values suggested that to obtain a maximum protein content under 80% amplitude conditions, the FE system required a lower temperature and shorter time than the FL system.

The total phenolic content ranged from 86.04 ± 1.92 to 127.84 ± 0.68 mg GAE/g for FE and from 108.75 ± 5.31 to 133.06 ± 1.25 mg GAE/g for FL, with slightly higher values for extracts pretreated by freeze-drying observed in all cases (Figure 3c). These results support the hypothesis that the freeze-drying treatment is more effective in preserving antioxidant compounds against oxidation in fruit waste [12]. The maximum values for this parameter corresponded to samples 13 and 14 for FE (central point, 50% amplitude, 50 °C, and 8 min) and to samples 1, 2, 4, 12, 13, and 15 for FL. Thus, a plateau of points and experimental conditions was involved in obtaining the maximum total phenolic content. Sample 6 (20% amplitude, 50 °C, and 12 min) showed the minimum value for FE, whereas sample 5 (80% amplitude, 50˚C, and 12 min) provided the lowest magnitude for FL. The obtained magnitudes were in the range of those previously found for eggplant skins (102.89 ± 1.87 mg GAE/g) and were higher than those found for lime (45.00 ± 1.45 mg GAE/g) and papaya (47.98 ± 2.42 mg GAE/g) skins by Vargas y Vargas et al. [21]. Other authors [1] have reported lower total phenolic contents (mg GAE/g) for lime (74.80), orange (66.36), and tangerine (58.68) skins treated under bath sonication conditions using water as solvent (30 min, 40 °C). Other extraction methods, such as microwave-assisted extraction using ethanol as solvent (60%), were able to recover lower total phenolic contents (mg GAE/g) from apple pomace (15.80) and citrus peel (23.63) [16].

The antioxidant capacity variation of the above samples is presented in Figure 3d, with similar tendencies observed for extracts from FE and FL samples. It should be highlighted that samples 5 (30 °C) and 15 (50 °C) presented higher antioxidant capacity values for extracts from conventionally dried (FE) fruit waste than for lyophilized (FL) waste, again suggesting the influence of the sample pretreatment conditions on the characteristics of the soluble extracts. The highest values for this parameter (mg Trolox/g) for both FE (53.10 ± 2.14) and FL (53.95 ± 1.28) were identified for sample 4 (50% amplitude, 70 °C, 4 min), whereas the lowest values (FE: 38.08 ± 3.03, FL: 39.44 ± 2.30) were obtained for sample 1 (50% amplitude, 70˚C, 12 min). No statistical differences were observed between samples 1, 2, and 3. Note here that higher antioxidant capacity values were identified for shorter sonication durations at a fixed temperature and amplitude. Indeed, the antioxidant capacity values (mg Trolox/g) achieved for both FE and FL samples were higher than those previously reported for mango skin (23.27 ± 62 0.23) and grape seeds (23.15 ± 0.09) using conventional extraction with organic solvents [7]. The freeze-dried by-products of lemon (33.17 ± 2.94 mg TEAC/g), orange (20.13 ± 0.43 mg TEAC/g), and apple (25.69 ± 0.56 mg TEAC/g) treated using ethanol as the extractive agent also provided lower values [13]. These results emphasize the relevance of studying the extraction conditions and reinforce the suitability of ultrasonic-probe-assisted extraction. Note here that the total phenolic content of the FL samples was systematically slightly higher than that of the FE samples (Figure 3c); however, the antioxidant capacity of the FL samples was not higher than that of the FE samples. This behavior suggests that it was not only the phenolic content that influenced the antioxidant capacity of the tested extracts. Nevertheless, further studies of the antioxidant capacity of these samples should be carried out to corroborate these outcomes.

### 3.4. Optimized Extraction Conditions

In order to obtain the maximum extraction yield through ultrasonic-probe-assisted extraction without jeopardizing the bioactive properties of the soluble extracts, the impact of the most relevant operational variables was assessed. Table 3 summarizes the regression coefficients and the corresponding statistical analysis required to define the objective functions for extracts from FE and FL samples. The objective function for the extraction yield, described using the coefficients with a significance degree >95%, can be expressed as:Yield FE = 46.57 − 0.023 T + 0.28 t(1)
Yield FL = 46.17 + 0.209 T − 0.25 t(2)

The representative response surface curves plotted in Figure 4a show the evolution of the extraction yield with the temperature and time at a fixed amplitude (50%). It can be observed that the profiles barely varied for short extraction durations with an increasing temperature, whereas from 50 °C, the yield increased as the temperature and extraction time rose, exhibiting a more pronounced effect at higher temperatures and longer durations. A linear and positive effect of temperature and time was identified, with higher yield values reached at higher temperatures and longer durations. The maximum yield predicted by the model (FE: 56.23%, FL: 56.17%) for both FE and FL samples corresponded to the conditions of 80% amplitude, 70 °C, and 12 min. These data corroborated the trends observed experimentally and are in agreement with the behaviors previously found for other fruit skins [17,33] treated by ultrasound.

Concerning proteins (mg/g), the objective functions (significance degree >95%) can be expressed according to Equations (3) and (4):Protein FE = −0.044 + 0.0077 A + 0.0073 T(3)
Protein FL = 0.060 − 0.0087 A + 0.0085 T + 0.021 t + 0.000086 A2 (4)

The selected response surface curves plotted in Figure 4b present the behavior of the protein content according to changes in the temperature and amplitude at fixed durations (8 min). For both FE and FL samples, higher temperatures and amplitudes suggested a higher protein content. The maximum value for the protein content (g/100 g) predicted by the model for FE (around 0.49) was observed under the conditions of 20% amplitude, 70˚C, and 12 min, whereas the model provided a maximum protein content (about 0.69) at 80% amplitude, 70 °C, and 4 min for FL.

The total phenolic content (mg GAE/g) exhibited a significant impact (>95%) from the quadratic terms of the objective function, as follows:Total phenolic content FE = 28.4 − 0.0156 A2 − 0.56 t2(5)
Total phenolic content FL = 108.4 − 0.0087 A2(6)

Figure 4c shows the representative response surface curves for extracts from FE and FL samples in terms of amplitude and time at a fixed temperature (50 °C). The optimum value for the total phenolic content (mg GAE/g) was identified as around 125.06 for FE (55.1% amplitude, 45.8 °C, and 8.3 min) and about 136.7 for FL (67.3% amplitude, 70 °C, and 4 min). Fierascu et al. [12] also indicated that the amount of antioxidants obtained from dried mango peel increased after freeze-drying when compared with conventional procedures. The importance of the extraction time for the total phenolic content has been described for other fruit skins [17,34]. The authors emphasized that these compounds are released in the initial period of extraction and increase over time, but that they decrease over longer periods of time due to the continuous collapse of the microbubbles that induce the degradation of the compounds.

The antioxidant capacity (mg Trolox/g) was significantly modified by temperature and time:Antioxidant capacity FE = 51.1 + 0.54 T − 2.7 t (7)
Antioxidant capacity FL = 52.37 − 2.05 t − 0.049 T t + 0.20 t2(8)

Figure 4d displays the corresponding response surface curves for the FE and FL systems. For both systems, a shorter extraction duration led to higher antioxidant capacity values at any amplitude, except for FL samples at amplitude values lower than 40%. The maximum antioxidant capacity predicted by the model for FE samples (53.66 mg Trolox/g) was observed under the conditions of 36.36% amplitude, 45.35 °C, and 4 min, whereas the maximum value for FL samples (54.40 mg Trolox/g) was predicted at 20% amplitude, 70 °C, and 4 min.

The joint optimization of all studied variables with a significance degree > 95% predicted that the best extraction conditions, which would achieve the highest yield (52.6%) without jeopardizing the protein content (0.42 mg/g), total phenolic content (116.42 mg GAE/g), and antioxidant capacity (44.95 mg Trolox/g), were 69.7% amplitude, 53.43 °C, and 12 min for FE. Similarly, the highest predicted parameters for FL (yield: 53.95%, protein content: 0.69 mg/g, total phenolic content: 135.32 mg GAE/g, and antioxidant capacity: 49.52 mg Trolox/g) involved a higher amplitude (80%) and temperature (70 °C) and a shorter extraction time (4 min).

## 4. Conclusions

Fruit industrial processing waste provides an alternative, economical, and natural source for valuable substrates obtained using appropriated technologies and procedures. Fruit waste pretreated by convective air drying required a three times longer extraction duration, though with a 15% lower ultrasound amplitude and a 25% lower extraction temperature, to achieve comparable extraction yields (around 50%). The tested properties were enhanced for the lyophilized samples, although twice the dehydration time and a higher energy consumption were required compared to those treated by conventional drying. The proposed green ultrasound treatment is a simple and flexible method that allows the quick extraction of high-value compounds and offers an integral valorization technique for fruit waste that could be extensible to other wastes. Further antioxidant capacity analysis is required to establish clear relationships between the total phenolic content and antioxidant potential of the extracts. Overall, the obtained soluble extracts of these fruit bioresidues open up a wide range of possible applications, including the enrichment and development of new day-to-day foods with beneficial properties, taking into account the promotion of health and wellbeing, and the creation of active packaging, biodegradable polymer packaging, and film wrapping, providing new prospects for the food market within the concept of a circular economy.

## Figures and Tables

**Figure 1 foods-11-02089-f001:**
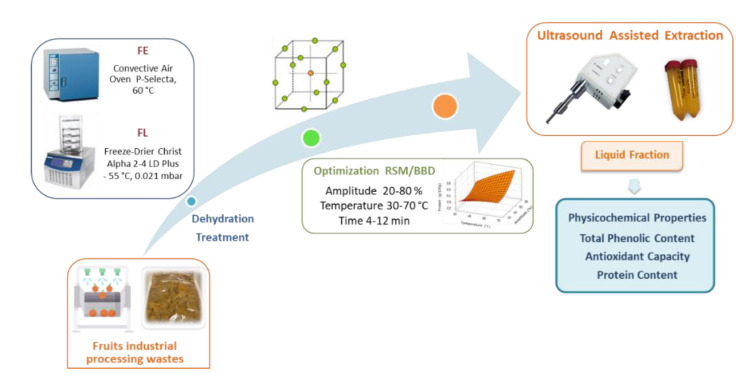
General scheme of the fruit waste extraction process.

**Figure 2 foods-11-02089-f002:**
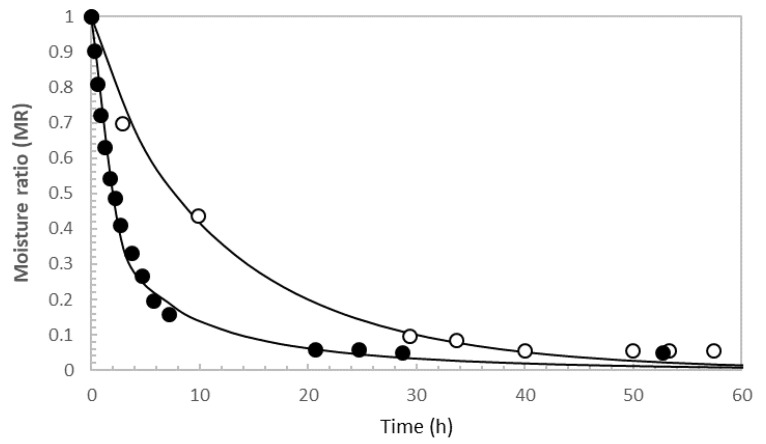
Drying kinetics of fruit industrial processing waste dehydrated using conventional (closed circles) and freeze-drying (opens circles) methods. Lines correspond to the Page model. Error bars smaller than symbol sizes.

**Figure 3 foods-11-02089-f003:**
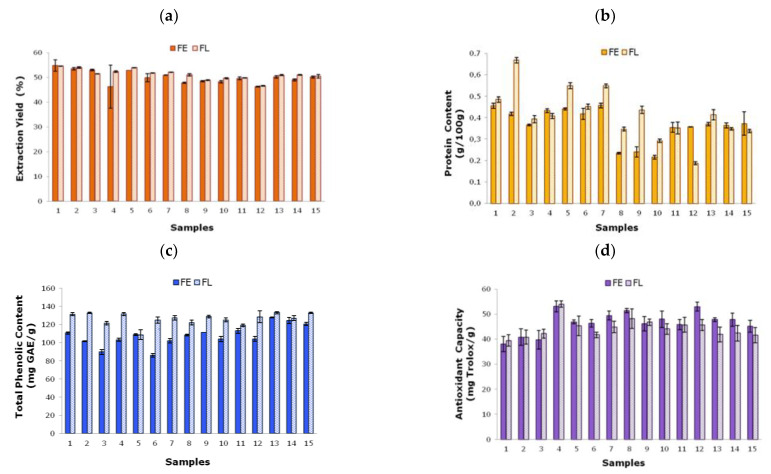
Experimental data of soluble extracts obtained after ultrasound-assisted extraction under different processing conditions (Table 1) for FE and FL samples: (**a**) yield, (**b**) protein content, (**c**) total phenolic content and (**d**) antioxidant capacity.

**Figure 4 foods-11-02089-f004:**
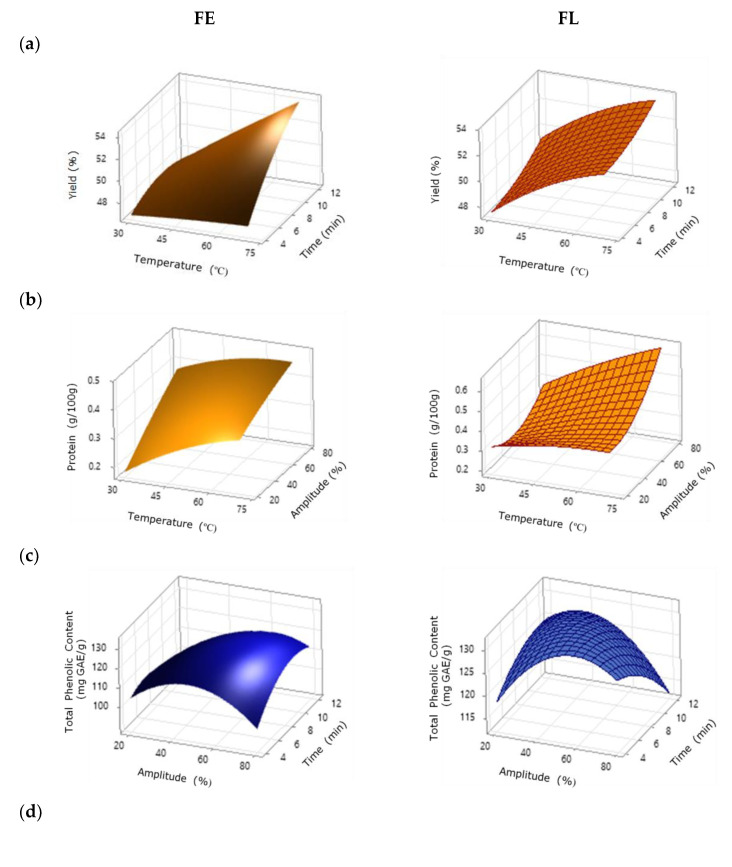
Selected response surface plots of (**a**) extraction yield (amplitude 50%), (**b**) protein content (time 8 min), (**c**) total phenolic content (temperature 50 °C), and (**d**) antioxidant activity (temperature 50 °C) as a function of the tested factors for the soluble extracts from dehydrated fruit waste. Note here that data in brackets correspond to the fixed value.

**Table 1 foods-11-02089-t001:** Box–Behnken experimental design for tested fruit waste, expressed in terms of dimensional and dimensionless independent variables.

Trials	Variables (Coded Levels)		
	A, % (X_1_)	T, °C (X_2_)	t, min (X_3_)
1	50 (0)	70 (0)	12 (0)
2	80 (1)	70 (0)	8 (−1)
3	20 (0)	70 (1)	8 (−1)
4	50 (0)	70 (1)	4 (1)
5	80 (−1)	50 (0)	12 (1)
6	20 (0)	50 (−1)	12 (1)
7	80 (0)	50 (0)	4 (0)
8	20 (−1)	50 (0)	4 (−1)
9	50 (0)	30 (0)	12 (0)
10	20 (−1)	30 (1)	8 (0)
11	80 (0)	30 (−1)	8 (−1)
12	50 (1)	30 (−1)	4 (0)
13	50 (1)	50 (1)	8 (0)
14	50 (1)	50 (0)	8 (1)
15	50 (−1)	50 (−1)	8 (0)

**Table 2 foods-11-02089-t002:** Fundamental composition of the dehydrated fruit waste (FE: conventionally dried and FL: lyophilized) used as raw material.

	FE	FL
**Moisture content (%)**	8.17 ± 0.18 ^a^	6.22 ± 0.24 ^b^
**Ash content (%)**	3.05 ± 0.34 ^a^	2.81 ± 0.17 ^a^
**Total protein content (%)**	6.32 ± 0.49 ^a^	5.34 ± 0.09 ^b^
**Carbon content (%)**	45.51 ± 0.18 ^a^	47.43 ± 2.87 ^a^
**Carbohydrate content (%)**	49.00 ± 0.86 ^a^	51.13 ± 0.62 ^a^
**Macroelements (mg/kg)**		
Ca	4911 ± 58 ^a^	4345 ± 49 ^b^
K	7941 ± 65 ^a^	7344 ± 71 ^b^
Mg	919 ± 21 ^a^	817 ± 12 ^b^
Na	50.0 ± 9 ^a^	9.71 ± 0.86 ^b^
**Microelements (mg/kg)**		
Cu	9.46 ± 0.56 ^a^	3.44 ± 0.12 ^b^
Fe	24.46 ± 1.21 ^a^	15.13 ± 0.29 ^b^
Zn	7.97 ± 0.13 ^a^	7.87 ± 0.15 ^a^
Cd	<0.10	<0.10
Hg	<0.020	<0.020
Pb	0.014 ± 0.001 ^b^	0.038 ± 0.002 ^a^

Data are given as mean ± standard deviation. Data values in a row with different superscript letters are statically different (*p* ≤ 0.05).

**Table 3 foods-11-02089-t003:** Coefficients determined for the proposed models for fruit waste (FE: conventionally dried and FL: lyophilized) and the corresponding statistical parameters.

Coefficient	Yield (%)		Protein Content (mg/g)		Total Phenolic Content (mg GAE/g)		Antioxidant Capacity (mg Trolox/g)	
	FE	FL	FE	FL	FE	FL	FE	FL
*a*_0_ (%)	46.57	46.17	−0.116	0.060	28.4	108.4	51.1	51.9
*a*_1_ (%/%)	−0.095	−0.155	0.0084 ^a^	−0.0087 ^a^	1.17	0.85	0.017	−0.041
*a*_2_ (%/°C)	−0.023 ^a^	0.209 ^a^	0.0073	0.0085 ^a^	1.76	−0.61	0.54 ^a^	0.13
*a*_3_ (%/min)	0.28 ^a^	−0.25 ^a^	−0.019	0.021 ^a^	5.71	4.20	−2.70 ^a^	−2.29 ^a^
*a*_4_ (%/(% °C))	−0.00039	0.00098	−0.000036	0.000090	0.0011	0.0071	0.0013	−0.0013
*a*_5_ (%/% min)	−0.00020	0.0020	−0.00041	−0.00021	0.061	−0.045	0.0051	0.0146
*a*_6_ (%/°C min)	0.019	−0.00027	0.00043	−0.00054	0.0024	−0.0015	−0.026	−0.049
*a*_7 (_%/%^2^)	0.0015	0.0012	−0.000008	0.000086 ^a^	−0.0156 ^a^	−0.0087 ^a^	−0.00135	−0.00010
*a*_8 (_%/°C^2^)	−0.00019	−0.0015	−0.000058	−0.00044	−0.0201	0.0036	−0.0053	0.0033
*a*_9 (_%/min^2^)	−0.049	0.024	0.0016	0.0019	−0.56 ^a^	−0.15	0.176	0.197
F	6.56	24.25	557.53	1.73	4.97	1.97	4.16	148.92
R^2^	**89.48**	**92.30**	**78.52**	**93.56**	**88.30**	**84.79**	**87.77**	**78.35**

^a^ Coefficients significant at *p* > 95%. Note here that coefficient units provided in the table are related to the yield; for the other variables studied they would be a_0_ (%), *a*_1_ (mg/g %), *a*_2_ (mg/g °C), *a*_3_ (mg/g min), *a*_4_ (mg/(g % °C)), *a*_5_ (mg/(g % min)), *a*_6_ (mg/(g °C min)), *a*_7_ (mg/(g %^2^)), *a*_8_ (mg/(g °C^2^)), and *a*_9_ (mg/(g min^2^)).

## Data Availability

Data are contained within the article.

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
