# Peer review of "Ultrasound-Assisted Extraction of High-Value Fractions from Fruit Industrial Processing Waste"

_foods, 2022, doi:10.3390/foods11142089_

Round 1
Reviewer 1 Report
This work discussed ultrasound assisted extraction of high valuable fractions from fruit industrial processing wastes. There are some issues in this manuscript that should be addressed as follows:
· The meaning of the abbreviations should be clearly defined at their first mention.
· A conclusive statement should be mentioned at the end of “Abstract”.
· The novel points of this study should be addressed in the “Introduction” because there are previous reports that discussed a similar topic https://pubmed.ncbi.nlm.nih.gov/32920300/
· The research ethics committee that approved this study should be mentioned with the number of the approval code.
· Page 3: A reference for the “Box–Behnken experimental design” should be added.
· I think it is better to write each of the “Results” and “Discussion” into separate sections.
· A collective diagram summarizing the main findings of this study is recommended.
· The number of Conclusion heading “i.e. 5“ should be changed to “4”.
· The “Conclusion” should be summarized and modified to delineate the possible practical applications of the findings of the present study in the daily life.
· The manuscript should be revised by English-naïve speaker to improve the quality of the language.
· The manuscript should be checked regarding the grammatical errors and plagiarism.
Author Response
Reviewer 1
This work discussed ultrasound assisted extraction of high valuable fractions from fruit industrial processing wastes. There are some issues in this manuscript that should be addressed as follows:
The meaning of the abbreviations should be clearly defined at their first mention.
It was carefully revised. Please see as e.g. on lines 82-86 “Conventional drying was conducted on a convective air oven (P-Selecta, Barcelona, Spain) at 60 ºC until constant weight following the corresponding drying kinetics, samples were labelled as FE. Freeze-drying was carried out in freeze-drier (Alpha 2-5 LD plus, Martin Christ, Germany) operating at -55 ºC, 0.021 mbar, samples were labelled as FL.”
A conclusive statement should be mentioned at the end of “Abstract”.
It was included following your advice. Please see on lines 21-24 “Tested properties were enhanced for samples previously lyophilized although twice the dehydration time and higher energy consumption was required when compared with those previously treated under conventional drying”
The novel points of this study should be addressed in the “Introduction” because there are previous reports that discussed a similar topic https://pubmed.ncbi.nlm.nih.gov/32920300/
It was clarified. The indicated reference is a comprehensive review paper which has already been included in the manuscript, where the effect of a conventional ultrasound extraction is discussed for a number of different fruit wastes. Our experimental work is focused on the valorization of different fruit industrial wastes throughout the optimization of an environmentally friendly extraction process to obtain high valuable extracts, as-sessing the impact of the dehydration pre-treatment of the wastes.
The research ethics committee that approved this study should be mentioned with the number of the approval code.
We have not used biological samples, not animals not volunteers, only an industrial fruit waste.
Page 3: A reference for the “Box–Behnken experimental design” should be added.
It was included.
I think it is better to write each of the “Results” and “Discussion” into separate sections.
Thank you for your suggestion, authors prefer to keep both jointly to avoid duplication of content.
A collective diagram summarizing the main findings of this study is recommended.
Figure 1 was revised in order to provide a general overview of the developed work.
The number of Conclusion heading “i.e. 5“ should be changed to “4”.
It was amended.
The “Conclusion” should be summarized and modified to delineate the possible practical applications of the findings of the present study in the daily life.
It was carefully revised. Please see on lines 429-433 “Overall, the obtained soluble extracts of these fruit bioresidues open up a wide range of possible applications that range from enriching and developing new daily foods with beneficial properties, taking into account the promotion of health and well-being, to the creation of active packaging, biodegradable polymer packaging or film wrapping, creating new perspectives for the food market within the concept of circular economy.”
The manuscript should be revised by English-naïve speaker to improve the quality of the language.
English language was carefully revised throughout the whole manuscript.
The manuscript should be checked regarding the grammatical errors and plagiarism.
It was carefully revised.
Thank you very much for your helpful comments and suggestions
Reviewer 2 Report
Review for foods-1813506 and manuscript entitled: “Ultrasound assisted extraction of high valuable fractions from fruit industrial processing wastes”. This manuscript describes that fruit wastes provide a valuable substrate through the use of appropriate technologies and procedures. The manuscript is suitable for the journal and the subject fits well into the aims and scope of the journal. On the whole, the manuscript is interesting and provides useful results. In my opinion, improvements are mentioned below.
Detailed suggestions:
1.Could you explain why the Ingredient content was different between FE and FL?
2.These residues were predominantly peels, seeds, peduncles and remains of pulp of orange, banana, pear and apple in variable proportion. Could you illustrate the correct ratio?
3.Could you explain why Sodium values of tested wastes had greater difference between FE and FL? (50.0 and 9.71 for FE and FL respectively)
4.The values of total phenolic content of FL observed in all cases slightly higher than FE (Figure 3c), but the antioxidation of FL doesn't have more effective than FE (Figure 3d). Could you explain the reason?
Author Response
Reviewer 2
Review for foods-1813506 and manuscript entitled: “Ultrasound assisted extraction of high valuable fractions from fruit industrial processing wastes”. This manuscript describes that fruit wastes provide a valuable substrate through the use of appropriate technologies and procedures. The manuscript is suitable for the journal and the subject fits well into the aims and scope of the journal. On the whole, the manuscript is interesting and provides useful results. In my opinion, improvements are mentioned below.
Thank you very much for your helpful comments and suggestions.
Detailed suggestions:
- Could you explain why the Ingredient content was different between FE and FL?
It is related to the different dehydration pre-treatment. It was explained in the manuscript. Please see on lines 228-246 “Table 2 summarizes the effect of drying treatment on the composition of fruit wastes (i.e. FE, FL), observing statistical differences in some fractions. FE exhibited slightly higher moisture content (about 8%) when compared with FL (about 6%), although in both cases lower than 10% which guarantees dehydration, inhibits microbial growth and increases the stability of the samples [29]. Similar values were identified for the ash content, being consistent with those previously reported for lime skins (5.84 ± 0.39%) or eggplants (9.21 ± 0.21) [21]. Average protein content values varied between (in dry weight) 6.32 % for FE and 5.34 % for FL, which agrees with those obtained for apple pomace (2.7-5.3%) [9], banana peel (6.02%) or orange peel (5.97-6.4%) [4,30]. The larger differences in mineral content between FE and FL were identified for sodium, suggesting that that smaller ions can be more easily entrained during the lyophilization process. Concerning carbohydrate con-tent, no significant differences were observed between fruit wastes subjected to different dehydration treatments, with values around 50%. This magnitude is over the range of those found for fruit residues such as bananas or cocoa (47.68 and 47.78%, respectively) [31]. Monosaccharide composition indicated that glucose was the main sugar in the tested industrial fruit residues, followed by xylose and arabinose. The corresponding values were 26.7 ± 0.35%, 8.53 ± 0.06% and 3.57 ± 0.04% for FE, and 27.68 ± 0.18%, 8.24 ± 0.05% and 3.51 ± 0.02% for FL, respectively. Carbohydrate and protein content makes this bio-waste an attractive source to be used in fermentation substrates of organic origin [20].”
- These residues were predominantly peels, seeds, peduncles and remains of pulp of orange, banana, pear and apple in variable proportion. Could you illustrate the correct ratio?
An estimation was included in the manuscript. Please see on lines 78-81 “Fruits industrial processing wastes used as raw material were kindly provided for a local company (Galifresh, Pontevedra, Spain). These residues were predominantly peels (≈ 55%), seeds (< 5%), peduncles (< 5%) and remains of pulp (≈35 %)of orange, banana, pear and apple in variable proportion. All samples were frozen at -18 ºC in air-tight plastic bags before dehydration treatment.”
3.Could you explain why Sodium values of tested wastes had greater difference between FE and FL? (50.0 and 9.71 for FE and FL respectively)
It was included. Please see on lines 236-238 “The larger differences in mineral content between FE and FL were identified for sodium, suggesting that that smaller ions can be more easily entrained during the lyophilization process.”
- The values of total phenolic content of FL observed in all cases slightly higher than FE (Figure 3c), but the antioxidation of FL doesn't have more effective than FE (Figure 3d). Could you explain the reason?
It was discussed. Please see on lines 339-343 “Note here that total phenolic content of FL was systematically slightly higher than FE samples (Figure 3c), however the antioxidant capacity of FL was not more effective than FE. This behavior suggests that not only phenolic content influence the antioxidant capacity of tested extracts. Anyway, further studies of antioxidant capacity should be developed with these samples to corroborate these outcomes.”

Reviewer 3 Report
The authors presented an interesting finding on extracting antioxidative components from the fruit industrial wastes. However, there are some questions that I would like to point out here before making a final decision.
1. May I know why the authors did not investigate the total carbohydrate content of soluble extracts obtained after ultrasound-assisted extraction of FE and FL (samples 1-15), as shown in Figure 3?
2. Secondly, why did the authors not do the correlation analysis between the antioxidant capacity and the total phenolic, protein, or carbohydrate content? This could provide valuable info to see which phytochemical content contributes to the total antioxidant capacity of the extracts.
3. The authors used the HPLC method to quantify monosaccharide content (Line 108-117). Is this a new method? If yes, did the author validate the method? If not, please provide the reference. Please include the chromatogram to show the separation of each monosaccharide and their retention time and compare the profile with a chosen extract.
Author Response
Reviewer 3
The authors presented an interesting finding on extracting antioxidative components from the fruit industrial wastes. However, there are some questions that I would like to point out here before making a final decision.
- May I know why the authors did not investigate the total carbohydrate content of soluble extracts obtained after ultrasound-assisted extraction of FE and FL (samples 1-15), as shown in Figure 3?
Carbohydrate content is one of the major fractions in fruits, which has been more studied in the literature. This study working with fruit industrial wastes pre-treated by two dehydration methods was focused on the potential of an environmentally friendly extraction process as ultrasound (by probe) for other fractions in terms of extraction yield, proteins and phenolic content, but we will take it in consideration for further works.
- Secondly, why did the authors not do the correlation analysis between the antioxidant capacity and the total phenolic, protein, or carbohydrate content? This could provide valuable info to see which phytochemical content contributes to the total antioxidant capacity of the extracts.
Thank you for drawing our attention on this relevant issue. The work was more focused on the extraction processing conditions. The joint optimization of all studied variables is provided in Figure 4. Note here that no other relevant correlations were identified.
- The authors used the HPLC method to quantify monosaccharide content (Line 108-117). Is this a new method? If yes, did the author validate the method? If not, please provide the reference. Please include the chromatogram to show the separation of each monosaccharide and their retention time and compare the profile with a chosen extract.
A reference was included. Please see on lines 112-115 “Carbohydrate profiles were determined by means of high-performance liquid chromatog-raphy (HPLC) with a previous acid hydrolysis treatment [22].”
As the paper is not focused in the carbohydrate fraction, and no supplementary Figures are allowed in Foods Journal, authors would prefer not to include simple HPLC profiles in the main manuscript. Anyway, we leave this decision the Editorial hands.
Thank you very much for your helpful comments and suggestions

Reviewer 4 Report
This manuscript was well written with an interesting topic about Ultrasound Assisted Extraction (UAE) of fruit-by-products from fruit industrial processing wastes. The experiments were carried out properly however, these points below should be asked to the authors.
Major concerns:
1. What is the main objective of this research? Would you like to compare the composition and antioxidation activity of FE and FL extracts after using UAE? Please consider the topic again because in this topic, it sounds like you want to compare UAE with conventional extraction method.
2. The conclusion should be revised by summarizing the details in the previous section. Please mention about antioxidation and total phenolic in the conclusion section.
Comments in details:
1. Please check reference format, especially ref 33.
2. Line 35 and 56; Please check the sentence in the parenthesis. It seems that the sentence is not finished yet.
3. What is the different of 15 samples in Figure 3?
4. What are the valuable fractions mean? Please explain in the introduction.
Author Response
Reviewer 4
This manuscript was well written with an interesting topic about Ultrasound Assisted Extraction (UAE) of fruit-by-products from fruit industrial processing wastes. The experiments were carried out properly however, these points below should be asked to the authors.
Thank you very much for your helpful comments and suggestions.
Major concerns:
- What is the main objective of this research? Would you like to compare the composition and antioxidation activity of FE and FL extracts after using UAE? Please consider the topic again because in this topic, it sounds like you want to compare UAE with conventional extraction method.
It was amended. The main objective was clarified. This study working with fruit industrial wastes pre-treated by two dehydration methods was focused on the potential of an environmentally friendly extraction process as ultrasound (by probe) for other fractions in terms of extraction yield, proteins and phenolic content.
- The conclusion should be revised by summarizing the details in the previous section. Please mention about antioxidation and total phenolic in the conclusion section.
It was revised following your suggestions. Please, see on lines 425-442 “Fruit industrial processing wastes provide an alternative and economical natural source for their transformation into a valuable substrate using appropriated technologies and procedures. Fruit wastes pre-treated by convective air drying required 3 times longer extraction times although with 15% less ultrasound amplitude and 25% less extraction temperature to achieve comparable extraction yields (around 50%). Tested properties were enhanced for samples previously lyophilized although twice the dehydration time and higher energy consumption was required when compared with those previously treated under conventional drying. The proposed green ultrasound treatment is a simple and flexible method that allows reaching extraction of high valuable compounds in short times and offering an integral valorization of fruit wastes that could be extensible to other wastes. Further antioxidant capacity analysis are required to stablish clear relationships between total phenolic content and antioxidant potential of the extracts. Overall, the ob-tained soluble extracts of these fruit bioresidues open up a wide range of possible applica-tions that range from enriching and developing new daily foods with beneficial proper-ties, taking into account the promotion of health and well-being, to the creation of active packaging, biodegradable polymer packaging or film wrapping, creating new perspec-tives for the food market within the concept of circular economy.”
Comments in details:
- Please check reference format, especially ref 33.
It was carefully revised.
- Line 35 and 56; Please check the sentence in the parenthesis. It seems that the sentence is not finished yet.
It was amended.
- What is the different of 15 samples in Figure 3?
It was clarified. Please see on lines 286-287 “Figure 3. Experimental data of soluble extracts obtained after ultrasound assisted extraction at different processing conditions (Table 1) of FE and FL.”
- What are the valuable fractions mean? Please explain in the introduction.
It was clarified in the introduction. Please, see on lines 32-36 “Its use for the production of high added value products, including biofertilizers, dietary fibers, industrial enzymes or biogas, as well as a substrate for the production of several valuable fractions as bioactive compounds (phenolic compounds, vitamins, polysaccharides , essential oils, among others), has caused great scientific interest in recent years [5].”
